# Reanalysis of the Sydney Harbor RiverCat Ferry

**Lawrence J. Doctors** 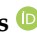

The University of New South Wales, Sydney 2052, Australia; L.Doctors@UNSW.edu.au

**Abstract:** In this paper, we revisit the hydrodynamics supporting the design and development of the RiverCat class of catamaran ferries operating in Sydney Harbor since 1991. More advanced software is used here. This software accounts for the hydrodynamics of the transom demisterns that experience partial or full ventilation, depending on the vessel speed. This ventilation gives rise to the hydrostatic drag, which adds to the total drag of the vessel. The presence of the transom also creates a hollow in the water. This hollow causes an effective hydrodynamic lengthening of the vessel, which leads to a reduction in the wave resistance. Hence, a detailed analysis is required in order to optimize the size of the transom. It is demonstrated that the drag of the vessel and the wave generation can be predicted with good accuracy. Finally, the software is also used to optimize the vessel further by means of affine transformations of the hull geometry.

**Keywords:** ferry design; wave generation; ship hydrodynamics

## 1. Introduction

### 1.1. Previous Studies

The principal features of the design philosophy behind the fleet of RiverCat ferries were described by Doctors, Renilson, Parker, and Hornsby [1] and Hornsby, Parker, Doctors, and Renilson [2]. These ferries were specifically designed in order to minimize the wave generation because of the requirement to limit the erosion of the banks of the Parramatta River along which these ferries operate. To this end, a total of ten different proposed designs were considered in the investigation.

As listed by Doctors, Renilson, Parker, and Hornsby [1] (Table 1), these designs consisted of three catamarans (same demihulls, but different demihull separations and displacements), a different set of three catamarans (same demihull, but with variations in the separations and displacement), and four trimarans (different subhull separations and different proportions of sidehull displacement relative to the centerhull displacement).

An elementary ship-resistance program was used in that study. This software complemented the purely experimental work of Renilson [3], in which the 1/25-scale models of the ten prospective ferries were tested in the towing tank at the Australian Maritime College (AMC). Both the physical experiments and the theoretical study suggested that the catamaran design was superior to the trimaran in terms of the wave generation, as characterized by the maximum height of the wave system at the specified speed and distance from the track of the vessel.

Two photographs of the RiverCat are shown in Figure 1. The general arrangements are shown in plan and profile in Figure 2. The extreme slenderness of the demihulls is very evident.

The particulars of the final design are listed in Table 1. It is particularly noteworthy that the demihull-beam-to-length ratio is 0.02857, which is likely to be a record for slenderness and is only possible due to the choice of a catamaran in order to solve the matter of lateral stability.

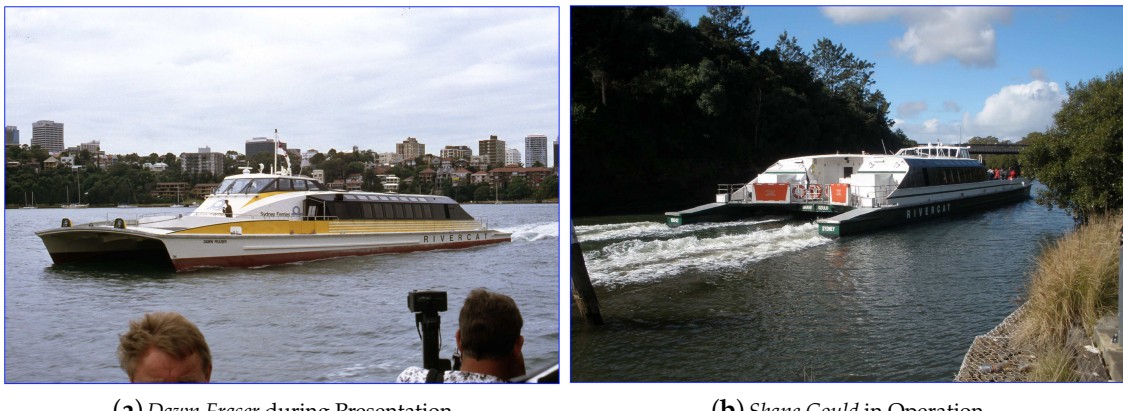

(**a**) *Dawn Fraser* during Presentation                    (**b**) *Shane Gould* in Operation

**Figure 1.** Grahame Parker Design Pty Ltd Sydney RiverCat.

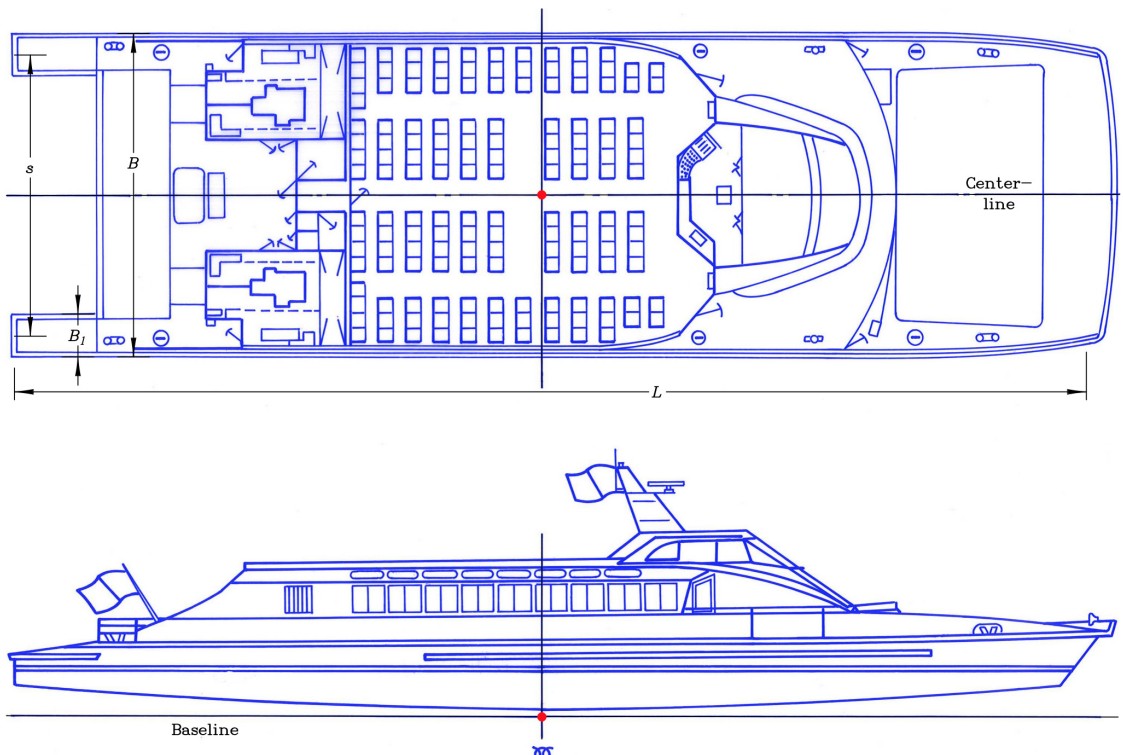

**Figure 2.** General arrangement of the Sydney RiverCat.

Because the key feature of the RiverCat is its very slender demihulls, it was not possible to position the propulsion engines inside them. As a consequence, the engines were placed on the deck behind the main passenger cabin. This may be seen in Figure 1b and in the plan view of Figure 2. More detail is presented in the profile view of the machinery in Figure 3 and in the section view of the machinery in Figure 4.

A negative feature of the unusual positioning of the engines is that the transmission had to be effected by means of steerable Z-drives, or azimuth thrusters, fitted with two right-angle gearboxes. These drives require more maintenance than conventional propeller shafts. On the other hand, there is a certain practical advantage of this arrangement because maintenance of the engines is made simple and very convenient.

**Table 1.** Particulars of the Sydney RiverCat.

| Quantity | Symbol * | Value |
|---|---|---|
| Length on waterline | $L$ | 35.00 m |
| Demihull beam | $B_1$ | 1.000 m |
| Beam overall | $B$ | 10.06 m |
| Draft | $T$ | 1.226 m |
| Block coefficient | $C_B$ | 0.6262 |
| Prismatic coefficient | $C_P$ | 0.6958 |
| Slenderness ratio | $L/\nabla^{1/3}$ | 11.68 |
| Transom–area ratio | $A_T/A_M$ | 0.4311 |
| Displacement mass | $\Delta$ | 55.00 t |
| Power | $P$ | $2 \times 335$ kW |
| Speed | $U$ | 23 kn |

* Nominal loading condition.

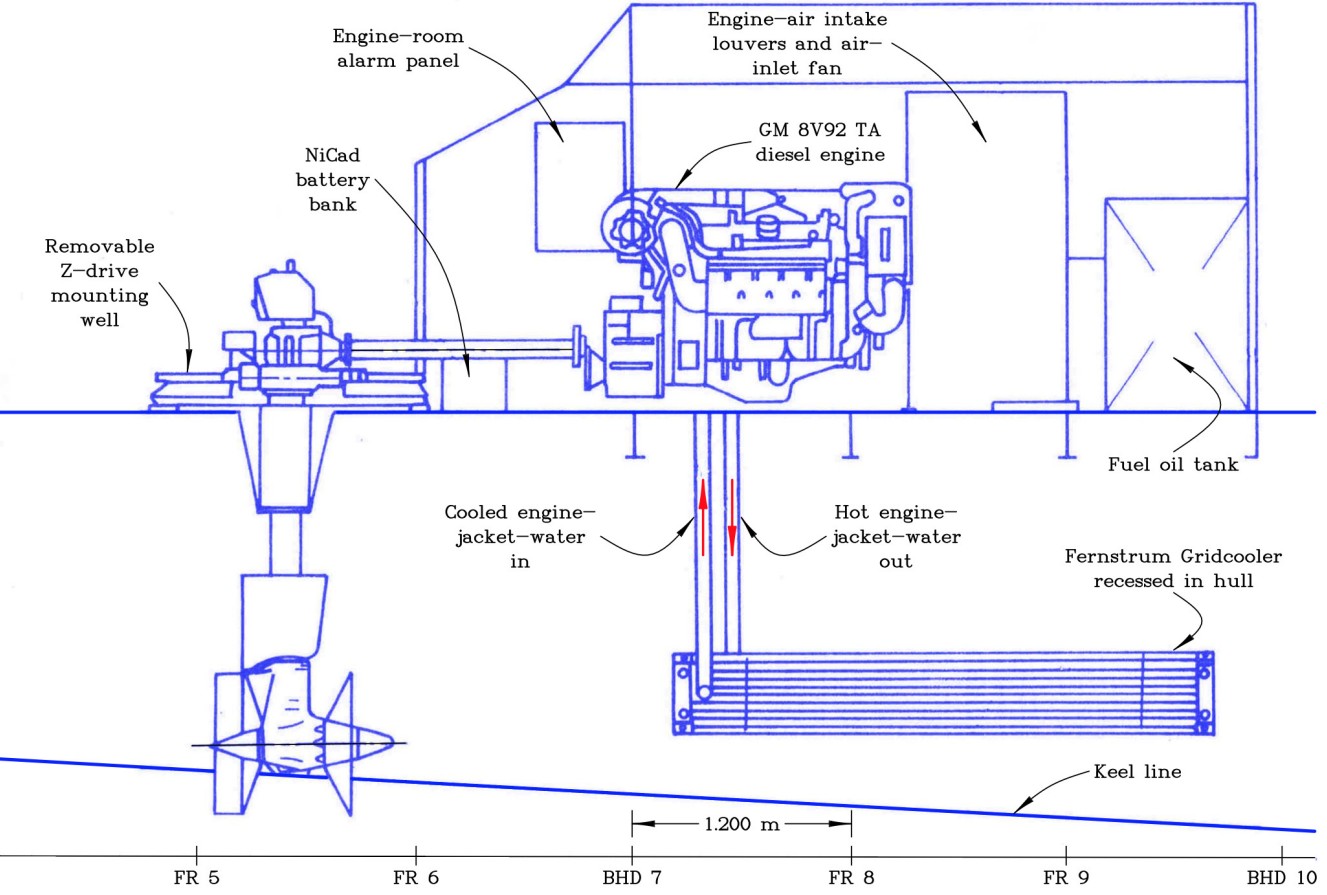

**Figure 3.** Profile of machinery arrangements.

Because of the low resistance of the vessel, other issues, such as cavitation and noise, have not been a practical concern. A significant design limitation was based on manning regulations. Thus, had the length of the vessel been increased slightly beyond the chosen value of 35 m, the required number of crew members would have been increased by one—leading to significantly increased operational costs.

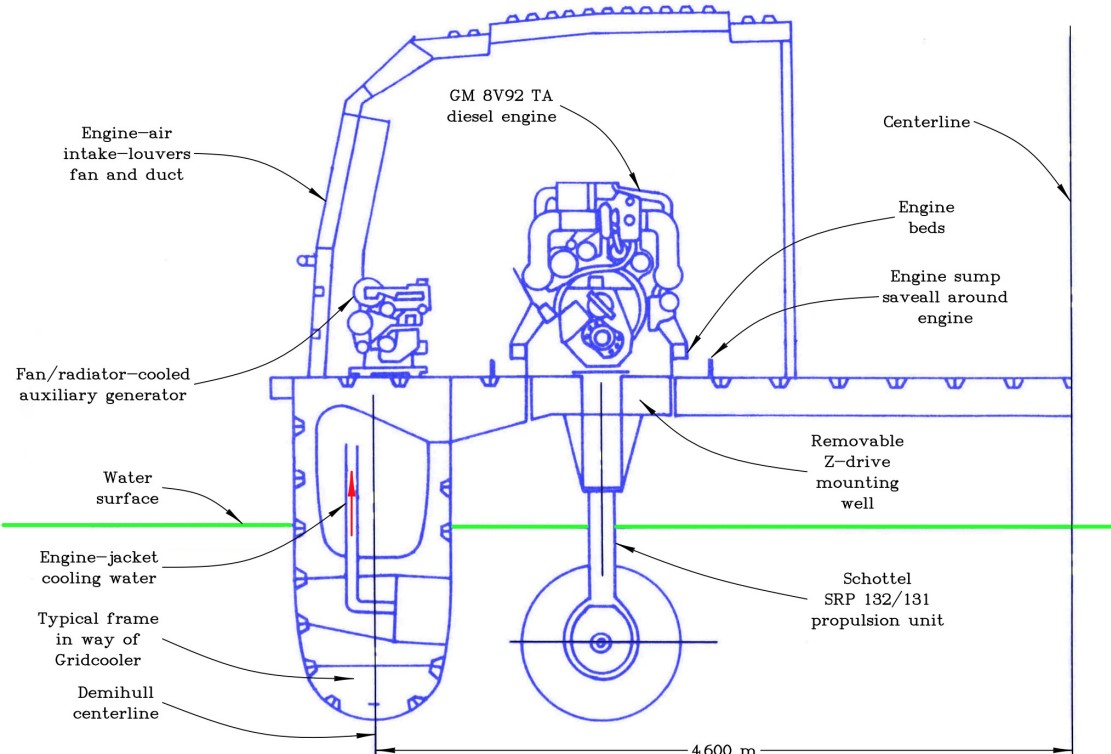

**Figure 4.** Section of machinery arrangements.

### 1.2. Current Investigation

The first purpose of the current study is to revisit the theoretical analysis supporting the design of this efficient ferry using more advanced software.

Because of the good correlation between this theory and the experimental data, the software is secondly applied to various modified geometries of the hull. These modifications will be effected by means of affine transformations of the individual demihulls.

There is a similarity between this second part of the study and the work of Doctors [4]. In that effort, vessels with one through six subhulls were analyzed. It was demonstrated theoretically that vessels with three or more subhulls experienced more wave resistance and more total resistance than the simpler catamaran. The higher total resistance of the multihulls relative to catamarans is readily explained by a consideration of their greater wetted surface, which creates more frictional resistance.

## 2. Hydrodynamic Theory

### 2.1. Decomposition of Resistance

We follow the traditional approach of decomposing the total resistance $R_T$ into reasonably independent components, as follows:

$$R_T = R_W + R_H + f_F R_F + R_A + R_a . \tag{1}$$

Here, $R_W$ is the wave resistance, $R_H$ is the hydrostatic resistance caused by the lack of hydrostatic pressure on the partly or fully ventilated transoms, $R_F$ is the frictional resistance, $R_A$ is the correlation allowance that accounts for roughness of the ship hull, and $R_a$ is the aerodynamic resistance, which will be neglected in this current effort. Lastly, $f_F$ is the frictional form factor, which accounts for the increased friction of a real vessel compared to that of a flat plate. This breakdown into components is similar to that proposed by Froude [5].



### 2.2. Wave Resistance

As the demihulls are considered to be thin, one may apply the classic potential-flow theory. The development of this theory can be traced to Michell [6], whose analysis applies to a monohull traveling in unrestricted water. The influence of laterally restricted water was included by Sretensky [7]. The effect of the finite depth of the water was added by Newman and Poole [8]. The extension of the analysis to a catamaran was presented by a number of researchers, including Doctors and Day [9]. A very detailed presentation of the theory was published by Doctors [10] (Section 5.2).

So, the theory accounts for the possible finite width and finite depth of the towing tank (in the case of a model) or the waterway (in the case of operation of the prototype in a river). The wave resistance is expressed as a sum of the effects of an infinite set of wave systems, each advancing at a different angle to the track of the vessel and at a different speed, such that each wave system keeps up with the vessel:

$$R_W = \frac{\rho g}{\pi} \sum_{i=0}^{\infty}{}' \epsilon \, \Delta k_y \, k k_x^2 \, (\mathcal{U}^2 + \mathcal{V}^2) \Big/ \frac{\mathrm{d}f}{\mathrm{d}k} \,, \tag{2}$$

$$\epsilon = \begin{cases} 1/2 & \text{for } i = 0 \\ 1 & \text{for } i \geq 1 \end{cases} \,. \tag{3}$$

The index $i$ for each wave component has been omitted from most of the algebraic expressions in order to simplify the notation. The other symbols are the water density $\rho$, the wave number $k$, the longitudinal wave number $k_x$, and the transverse wave number $k_y$. The prime $'$ on the summation in Equation (2) has been used to indicate that the zeroth term, which indicates the transverse wave, is to be omitted for the supercritical case. This is relevant when the depth Froude number $F_d = U/\sqrt{gd}$ exceeds unity, where $g$ is the acceleration due to gravity, $d$ is the depth of the water, and $U$ is the vessel speed.

The wave numbers are determined from the sequence of formulas:

$$\Delta k_y = 2\pi/w \,, \tag{4}$$

$$k_y = i \Delta k_y \,, \tag{5}$$

$$k_x^2 + k_y^2 = k^2 \,, \tag{6}$$

in which $w$ is the width of the channel and $k$ is the solution of the transcendental equation

$$f = k^2 - k_0 k \tanh(kd) - k_y^2 \,, \tag{7}$$

where $k_0 = g/U^2$ is the fundamental circular wave number. The solution of this implicit equation can be found in the usual way by using the Newton–Raphson iteration with the assistance of its derivative,

$$\mathrm{d}f/\mathrm{d}k = 2k - k_0 \tanh(kd) - k_0 k d \, \mathrm{sech}^2(kd) \,. \tag{8}$$

The two finite-depth-water Kochin functions in Equation (2) are defined by the formula

$$\mathcal{U} + \mathrm{i}\mathcal{V} = 2\cos\left(\tfrac{1}{2}k_y s\right) \int_{\mathcal{S}_1} b_1(x,z) \exp(\mathrm{i}k_x x) \frac{\cosh[k(z+d)]}{\cosh(kz)} \, \mathrm{d}\mathcal{S} \,, \tag{9}$$

in which $s$ is the separation between the demihull centerplanes, $b_1(x,z)$ is the local demihull beam, and $\mathcal{S}_1$ is the centerplane area of a demihull. A convenient location of the Cartesian coordinate origin is on the centerplane of the vessel at the stern on the undisturbed water surface, with $x$ directed forward, $y$ directed to port, and $z$ directed upward. The compu-

tation of the two functions in Equation (9) is conveniently related to the two deep-water Kochin functions,

$$\mathcal{P}^{\pm} + i\mathcal{Q}^{\pm} \quad = \quad 2\cos\left(\frac{1}{2}k_y s\right) \int\limits_{\mathcal{S}_1} b_1(x,z)\exp(ik_x x \pm kz)\,d\mathcal{S}\,, \tag{10}$$

by means of the pair of relationships:

$$\mathcal{U} \quad = \quad \frac{\mathcal{P}^+ + \exp(-2kd)\mathcal{P}^-}{1 + \exp(-2kd)}\,, \tag{11}$$

$$\mathcal{V} \quad = \quad \frac{\mathcal{Q}^+ + \exp(-2kd)\mathcal{Q}^-}{1 + \exp(-2kd)}\,. \tag{12}$$

### 2.3. Hydrostatic Resistance

The water pressure acting on the transom gives rise to a negative contribution to the force in the aft direction. Figure 5a is an idealization of the flow behind the transom in the partially ventilated condition. The presence of the deadwater region suggests that we can estimate the pressure load on the face of the transom by means of simple hydrostatic considerations.

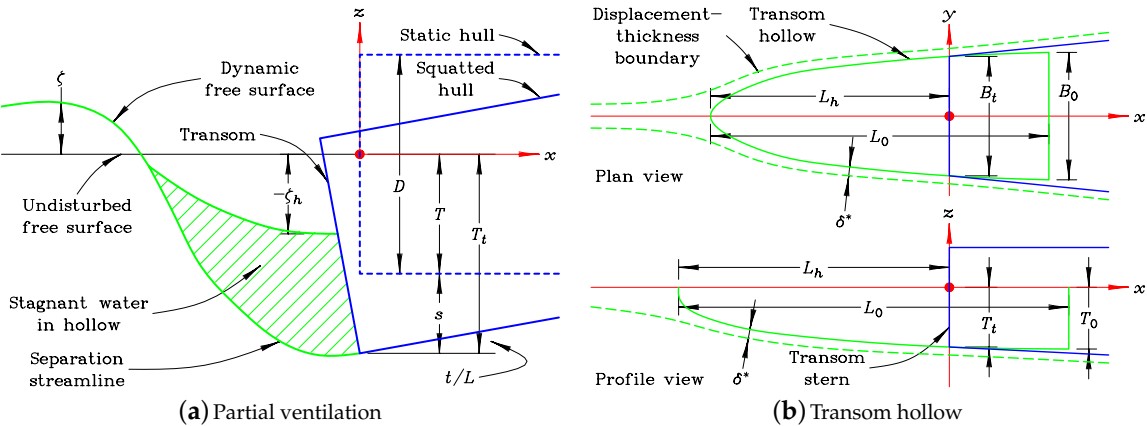

(**a**) Partial ventilation         (**b**) Transom hollow

**Figure 5.** Hydrodynamic modeling of the transom stern.

Therefore, this load is calculated by the simple integration:

$$R_{H1} \quad = \quad \rho g \int\limits_{-T_t}^{\zeta_t} b(x_t,z)(z - \zeta_t)\,dz\,, \tag{13}$$

where $T_t$ is the draft at the transom and $\zeta_t$ is the local elevation of the water surface on the face of the transom, which is always negative in this context.

On the other hand, if the transom were fully wetted, as in the at-rest condition, the hydrostatic force in the aft direction would be

$$R_{H2} \quad = \quad -\rho g \int\limits_{-T_t}^{0} b(x_t,z)z\,dz\,. \tag{14}$$

The final result for the drag is, therefore, a summation of these two contributions:

$$R_H \quad = \quad R_{H1} + R_{H2}\,. \tag{15}$$

### 2.4. Transom Hollow

The process of the ventilation of the transom has been researched by a number of workers. Some of the most practical design guidance was provided by Toby [11], Toby [12], and Toby [13]. As well as giving rise to the unwanted hydrostatic drag, the separation of the water flow at the stern creates a hollow in the water, which adds to the effective hydrodynamic wave-making length of the vessel. This hollow is illustrated in Figure 5b.

This occurrence of the transom hollow is generally favorable in that it reduces the wave resistance. The reader should consult Doctors [10] (Chapter 4) for an in-depth summary of the research on this question.

### 2.5. Frictional Resistance

The frictional resistance on the model will be computed by the use of the ITTC (1957) (International Towing-Tank Conference) formulation, described by Clements [14] (Page 374) and Lewis [15] (Section 3.5, Pages 7 to 15). This process first requires calculating the Reynolds number based on the wetted length $L$ of the vessel:

$$R_N = UL/\nu,\tag{16}$$

in which $\nu$ is the kinematic viscosity. The coefficient of frictional resistance is estimated for extrapolation purposes as

$$\begin{aligned}C_F &= R_F \Big/ \frac{1}{2}\rho U^2 S\\ &= 0.075/\left[\log(R_N)-2\right]^2,\end{aligned}\tag{17}$$

where $S$ is the area of the wetted surface of the vessel.

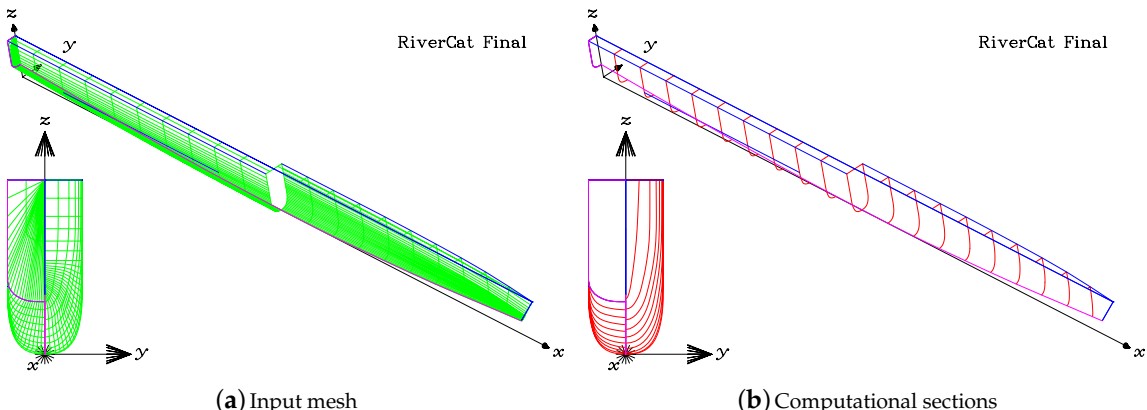

(**a**) Input mesh  (**b**) Computational sections

**Figure 6.** Demihull of the finalized RiverCat.

## 3. Characteristics of the RiverCat

### 3.1. General Layout

Figure 2 demonstrates the very slender nature of the demihulls of the vessel. This was a primary design feature with the specific purpose of minimizing the wave generation of the ferry. The choice of a catamaran permitted the selection of such slender hulls. Had a monohull design been chosen, it would not have been possible to achieve the minimum required transverse stability.

### 3.2. Demihulls

The input mesh defining the geometry of the demihull is reproduced in Figure 6a. For the purpose of the computations, a regular spacing of the transverse sections is prefer-

able. This reorganizing of the definition of the hull geometry is performed automatically by the software. The result of this process is shown in Figure 6b.

Both parts of Figure 6 show perspective split views of the demihull. Perhaps more useful are the body plans, or bow-on frontal views. These views emphasize the very rounded nature of the bilges, the very fine bow, and the transom. The pronounced rocker (rise of keel) gives as small a transom area as possible and is evident in this figure. The rocker is very noticeable in the profile view of Figure 2.

## 4. Numerical Computations

### 4.1. Finalized Vessel

Figure 7 shows the results of the towing-tank experiments and the current numerical analysis of the finalized design of the RiverCat. The calculations pertain to a model displacement of 3.434 kg. This corresponds to the prototype displacement of 55 t. The prototype demihull centerplane separation is 9.060 m in this example.

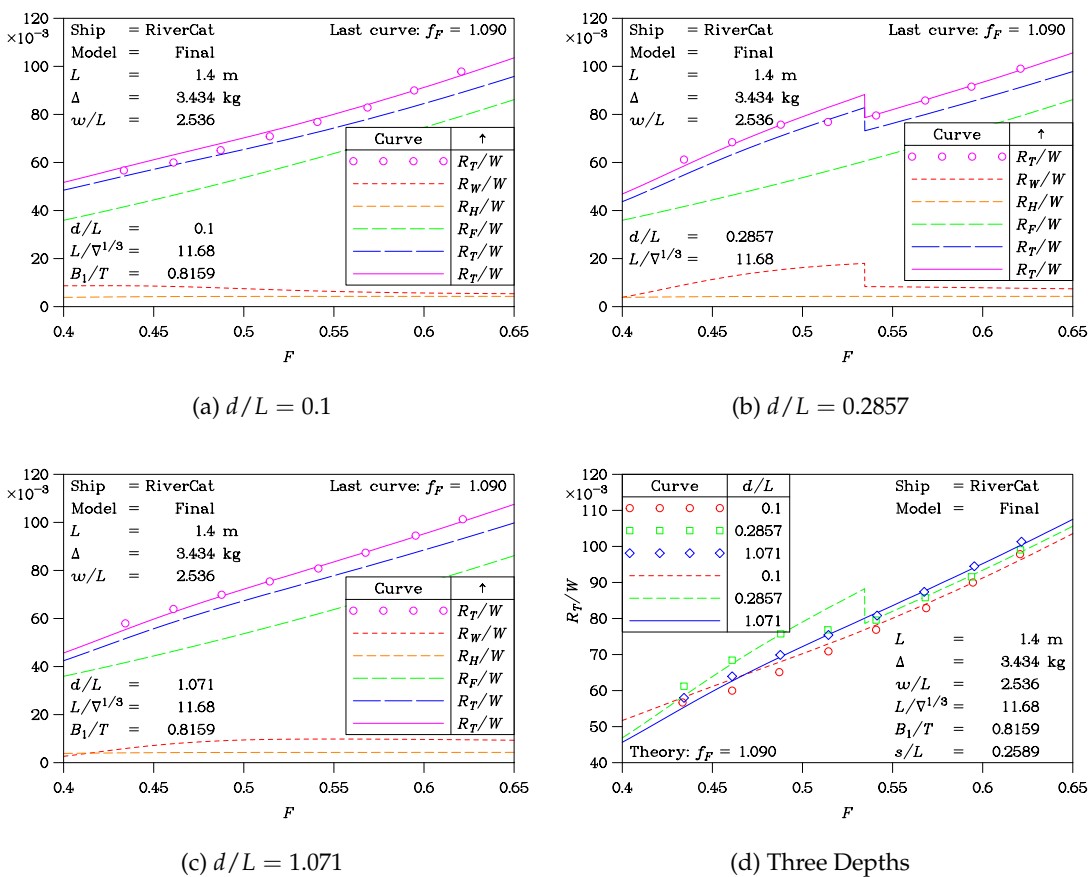

**Figure 7.** Experiments on the model of the Finalized RiverCat.

The RiverCat was designed to operate in very shallow water as well as in deep water. Thus, the first three plots in Figure 7 relate specifically to the depth-to-length ratios $d/L$ of 0.1, 0.2857, and 1.071. Each of these three plots presents the theoretically computed wave resistance $R_W$, the hydrostatic resistance $R_H$, and the ITTC (1957) frictional resistance $R_F$. The plotted data are non-dimensionalized with respect to the displacement weight $W$. So, these are the specific-resistance coefficients. These components are plotted as a function of the Froude number $F = U/\sqrt{gL}$.

It is imperative to emphasize the relatively small rôle that the wave resistance plays in contributing to the total resistance budget. Even in Figure 7b, for the dimensionless depth $d/L = 0.2857$, where the speed range covered by the experiments includes the critical speed defined by a unit value of the depth Froude number $F_d$, the lion's share of

the resistance is due to friction. On the other hand, it should be noted that in accordance with the principles of the Froude extrapolation, the resistance will be less important at the prototype scale because of the greater value of the Reynolds number.

The hydrostatic drag is evidently very small and is typically about one-half of the wave resistance—for speeds away from the critical value. The hydrostatic drag is also independent of the speed, at least for the range of speeds covered by these plots. This confirms that the transom is consistently fully ventilated in this speed range.

The plots in Figure 7 also show the total resistance $R_T$. Circular symbols have been used to indicate the experimental data points, and curves have been used to show the results of the computer predictions. The last dashed curve is the theoretical prediction according to the simple summation suggested by Equation (1)—with a unit value of the frictional-resistance form factor $f_F$.

In addition, the last and continuous curve in each case shows the total resistance, also using Equation (1), but with the more realistic choice $f_F = 1.090$. This value was deduced by means of a standard root-mean-square minimization procedure, such as that described by (de Vahl Davis [16] Section 3.10). The method was applied to Equation (1) with the one unknown $f_F$.

With this value of $f_F$, an excellent correlation between the predicted total resistance and the experimentally derived data is obtained. The most interesting case, depicted in Figure 7b, demonstrates the drop in resistance when the speed crosses the critical value. The steady-state resistance theory used here predicts a sharp drop, which is not so evident in the experiments.

The magnitude of this drop in specific resistance is given by the remarkably simple formula:

$$\Delta R_W / W \quad = \quad 3\nabla / 2wd^2 \,, \tag{18}$$

in which $\nabla$ is the displacement volume. This formula is independent of the shape of the vessel hull.

We can verify the magnitude of the drop in Figure 7b by noting that in this case, $\Delta = 3.434$ kg, $w = 3.550$ m, and $d = 0.400$ m. These data give $\Delta R_W / W = 9.069 \times 10^{-3}$; this result is in agreement with the theoretical discontinuities in the curves for the wave resistance and for the total resistance.

The fact that the discontinuity in the experimental data is somewhat rounded off—unlike the sharp discontinuity in the theory—can be resolved by employing the unsteady wave-resistance theory published by Day, Clelland, and Doctors [17]. It was demonstrated that the time-averaging signal processing, which is used in recording ship-model-resistance data during typical tests, results in a rounding of the results in the vicinity of $F_d \approx 1$. One must, therefore, also account for the motion of the tank carriage from rest and use the true time-unsteady theory. Then, the rounded characteristic of the resistance curve can be predicted accurately. An example for a case of $w/L = 1.524$ and $d/L = 0.25$ was published by (Day, Clelland, and Doctors [17] Figure 6).

Finally, we note that only the midrange of Froude numbers was of interest in this work. It is well known that the curve of wave resistance exhibits a large number of oscillations at low values of the Froude number. The reader is referred to Doctors [10] (Figure 5.11, Page 132), where the wave-resistance coefficient is plotted for Froude numbers down to zero in value. We add here that when the more useful specific wave resistance is plotted instead, the magnitude of these oscillations is greatly reduced, and so is their significance.

### 4.2. Transformations of the Finalized Demihull

In this section, we will consider four different types of transformations of the demihull of the Sydney RiverCat. These are listed in Table 2.

The first transformation involves stretching the length of the demihull and reducing the local beam and local draft in equal proportions, thus maintaining a fixed value of the displacement.

The second transformation relates to changing the local sectional aspect ratio $B_1/T$ while keeping the local sectional area and the length fixed.

The third transformation is to change the demihull separation by increasing it from the original value.

The fourth and the most challenging transformation is one in which the significance of the transom, as measured by the metric transom–area ratio $A_T/A_M$, is examined. In the current effort, the methodology of Doctors [4] is replicated. The first step is to create a pointed-stern version of the original demihull.

In this example, the forward half of the vessel is employed to create a pointed-stern demihull. This demihull is symmetric fore-and-aft. Of course, it is improper to directly compare the hydrodynamics of this demihull with the original transom-stern demihull because its volume is substantially lower. Consequently, the local beam and the local draft have each been increased at all stations in an affine manner by a simple factor, which is the square root of the desired ratio of volumes. The resulting pointed-stern demihull with the same displacement volume is presented in Figure 8a.

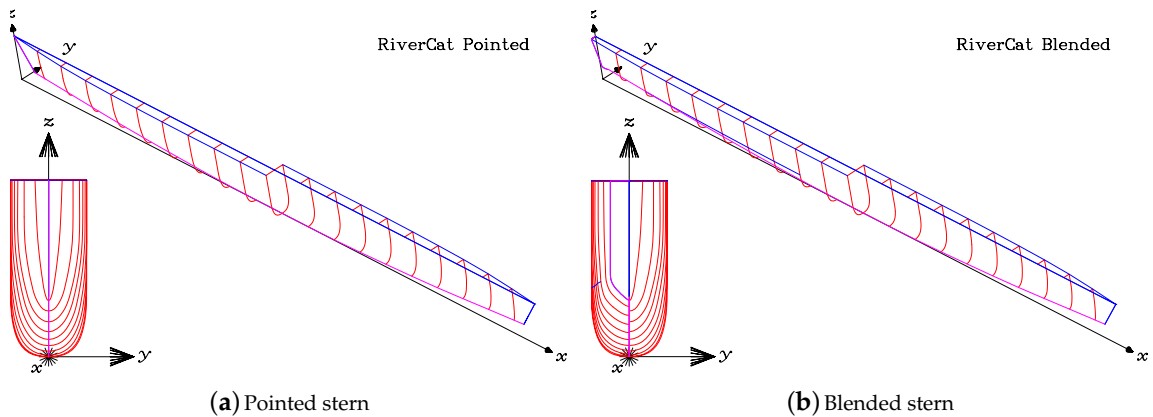

(**a**) Pointed stern        (**b**) Blended stern

**Figure 8.** Transformations of the finalized RiverCat demihull.

We now have two extreme cases. These are the original transom-stern demihull in Figure 6b and the pointed-stern demihull in Figure 8a. Next, a third and intermediate blended-hull demihull was created by combining or blending equal portions of these two basis demihulls, after the manner of Doctors [18]. That is, the coordinates of the points on the surface of the new and blended hull are simple averages of the original coordinates. The blended-stern vessel is presented in Figure 8b.

**Table 2.** Affine transformations of the demihull.

| Index | Parameter | Symbol | Values | | |
|---|---|---|---|---|---|
| | | | **Affine 0** [†] | **Affine 1** | **Affine 2** |
| 1 * | Slenderness ratio | $L/\nabla^{1/3}$ | 11.68 | 14.60 | 17.52 |
| 2 | Beam-to-draft ratio | $B_1/T$ | 0.8159 | 1.275 | 1.836 |
| 3 | Demihull separation | $s/L$ | 0.2589 | 0.3160 | 0.3731 |
| 4 | Transom–area ratio | $A_T/A_M$ | 0.4311 | 0.2156 | 0 |

* Corresponding to the four parts in each of Figures 9–11. [†] Data for the finalized RiverCat.

### 4.3. Wave Resistance

The wave generation of the Sydney RiverCat was a critical performance criterion in its design. To this end, we provide a set of calculations in each of the four parts of Figure 9.

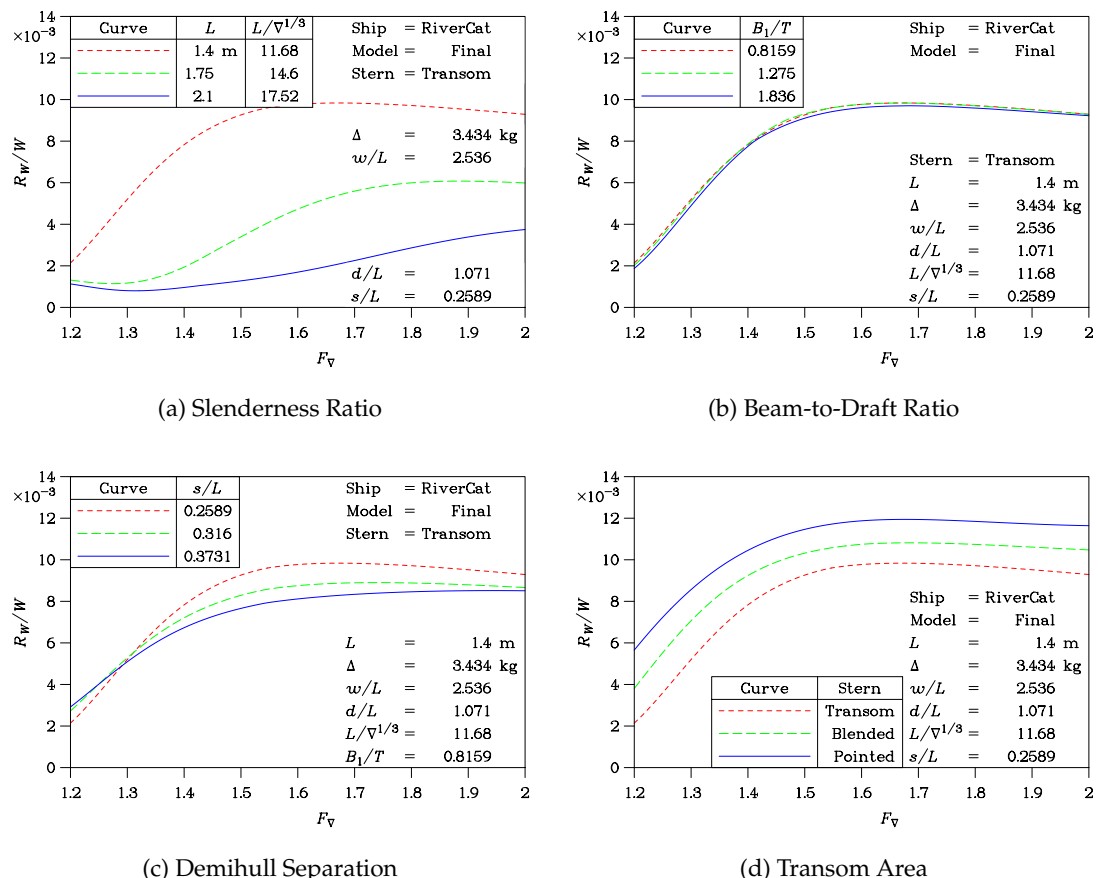

**Figure 9.** Influence of the affine transformations on wave resistance.

Figure 9a is a plot of the specific wave resistance $R_W/W$ as a function of the volumetric Froude number $F_\nabla = U/\sqrt{g\nabla^{1/3}}$. This abscissa was chosen instead of the traditional Froude number $F$ because the model length varies between the curves in the test of slenderness variation; so, it is advisable to be consistent and to use the displacement volume $\nabla$ as the basis for rendering the speed non-dimensional.

Three different values of the slenderness ratio $L/\nabla^{1/3}$ are considered in Figure 9a, varying between 11.68 for the finalized RiverCat and 17.52 for the longest variant. As noted earlier, the cross-sectional shape has been kept constant so that $B_1/T$ is the same for the three models. There is a significant reduction in wave generation for the slenderest vessel. This shows that there is much scope for reducing the undesired wave generation.

Modifying the section aspect ratio, as measured by the ratio $B_1/T$, is considered in Figure 9b. There is almost no influence of this parameter on wave generation.

The demihull–centerplane separation $s$ is changed in Figure 9c. Increasing the separation generally reduces the wave generation, as is well known from previous research. However, the effect is not large.

Lastly the importance of the type of vessel stern is examined in Figure 9d. The original and finalized RiverCat (with the full transom) is seen to experience the lowest wave resistance. This has been explained in the past as being a result of the transom hollow creating an effectively longer hydrodynamic shape in the water, which is favorable.

### 4.4. Model Total Resistance

The total resistance of the model is considered in the four plots of Figure 10. The variations in the demihull modifications studied in these plots correspond to the four parts of Figure 9.

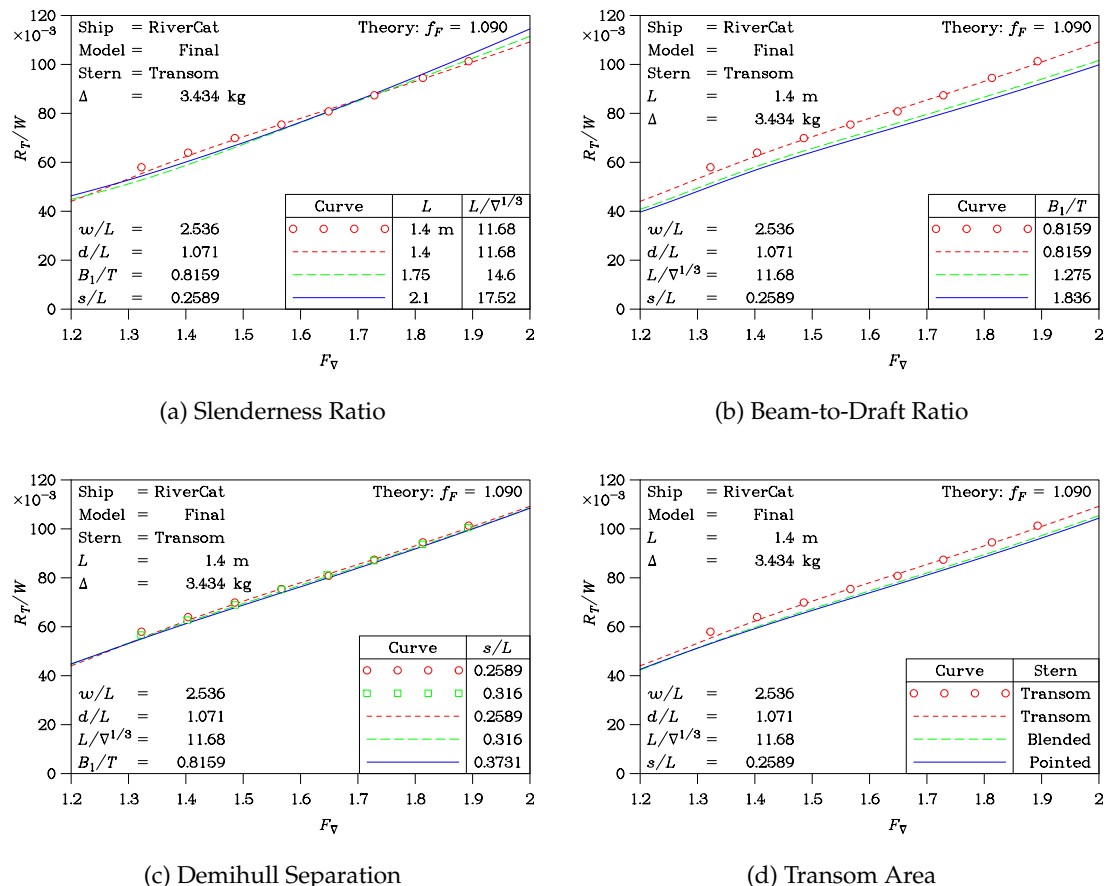

**Figure 10.** Influence of the affine transformations on model resistance.

The total resistance of the model was tested in some of these cases, and the deep-water model data for the finalized RiverCat were plotted already in Figure 7c.

The influence of slenderness in Figure 7a is seen to be negligible—despite its strong effect on wave resistance in Figure 9a. This outcome can be explained by the fact that slenderer hulls have a greater wetted-surface area, which is a negative attribute.

Increasing the section aspect ratio $B_1/T$ in Figure 7b is favorable because sections closer to semicircles have a smaller perimeter, leading to less wetted surface.

The effect of increasing the demihull separation in Figure 7c is almost negligible because, while the wave resistance is noticeably reduced, the lion's share of the resistance is due to frictional resistance, which is not altered.

Lastly, the influence of transom-stern size is considered in Figure 7d. Again, the reduced wave resistance for a demihull with a larger transom is not of great importance for the total resistance because it contributes so little to the total drag budget.

### 4.5. Prototype Total Resistance

Lastly, we emphasize that the main purpose of performing calculations on ship models is to verify the experimental data measured on physical models and to understand the hydrodynamic phenomena. Nevertheless, we are only concerned, in a more practical sense, with the performance of the prototype vessel. These theoretical results are presented in the four graphs of Figure 11, which correspond to the four graphs of Figure 10.

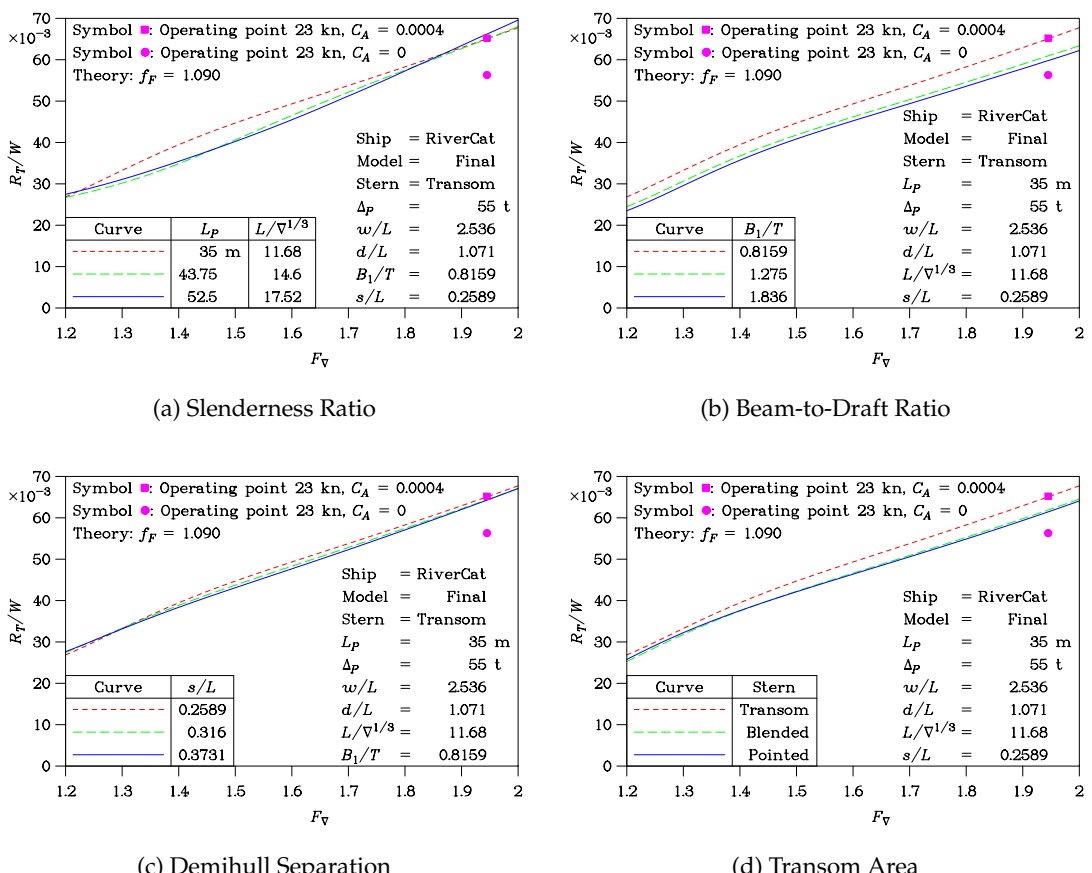

**Figure 11.** Influence of the affine transformations on prototype resistance.

To minimize the human effort required, these computations for the prototype were effected on the 1/25-scale hull geometry—as for the previous computations for the model. We remind the reader that the results are presented in a dimensionless manner. So, the only necessary change to the data file was to reduce the value of the kinematic viscosity $\nu$ in Equation (16) for the Reynolds number by a factor of $25^{3/2}$. In this manner, the software will calculate the correct Reynolds number at the prototype scale.

In broad terms, the plotted curves for the resistance of the prototype resemble those for the model. It is important to note that the upper limit for the vertical scale for the total resistance has been reduced from 0.12 to 0.07. That is, the values of the total resistance are almost one-half when plotted on a dimensionless basis. This substantial reduction in resistance is due to the much lower frictional-resistance coefficient, as computed using Equation (17). This is a fundamental outcome of Froude scaling. In line with standard practice, a correlation or roughness allowance $C_A = 0.0004$ has been added to the friction coefficient at prototype scale according to Equation (17).

Because the major part of the resistance is due to friction—even at the prototype scale— most of the variations in demihull geometry considered in Figure 11 are insignificant. This is particularly true at the nominal operating point, corresponding to a speed of 23 kn, which is indicated on the four plots.

### 4.6. Hull Finish

The transport factor, defined as

$$\text{TF} \quad = \quad WU/P \tag{19}$$
$$= \quad \eta \times (W/R_T), \tag{20}$$

is here evaluated on the basis of the total weight $W$, the operational speed $U$, and the installed engine power $P$. We now use the prototype displacement of 55 t, the nominal operating speed of 23 kn, and the prototype propulsive power of $2 \times 335$ kW. So, the transport factor TF is 9.524.

We also note that at this speed, corresponding to $F_\nabla = 1.945$, we have the predicted value of the specific-resistance data at the operating point from any of the four plots in Figure 11, namely, $R_T / W = 0.0652$. This allows us to use Equation (20) and to compute the overall propulsive efficiency of the transmission and the propellers $\eta$, which is 0.621. This result is a reasonably acceptable value in naval-architecture usage.

On the other hand, if the hull finish of the vessel could be maintained at an ideal hydraulically smooth value of zero, the transport factor rises to a value of 11.03. A strong case can be made for enforcing a strict régime for cleaning the hull on a regular basis, thereby reducing fuel consumption. These results are summarized in Table 3.

**Table 3.** Transport factor of the Sydney RiverCat.

| Assumption | Transport Factor TF * |
|---|---|
| ITTC (1957) with $C_A = 0$ | 11.03 |
| ITTC (1957) with $C_A = 0.0004$ | 9.52 |
| Measured | 9.52 |

* Nominal operating point: $\Delta$ = 55 t, $U$ = 23 kn and $\eta$ = 0.621.

Further studies on the matter of hull friction and different methods of implementing the ITTC (1957) extrapolation were published by Clements [14] (Page 374) and Lewis [15] (Section 3.5, Pages 7 to 15), the ITTC (1978) extrapolation by Oosterveld [19] (Equation (1.19)) and ITTC [20], and the ITTC (2004) extrapolation by Candries and Atlar [21] (Equation (1)) and ITTC [22] (Equation (3)).

## 5. Conclusions

### 5.1. Current Investigation

The current study has demonstrated that the linearized wave-resistance theory, used in conjunction with physical models of the transom-ventilation process and the generation of the accompanying transom hollow, provides excellent predictions of the resistance of efficient river catamarans. It is important, at the same time, to have good estimates of the frictional-resistance form factor. For the Sydney RiverCat, this was determined to be 1.09.

This work has also shown that it is difficult to further improve the hull design if one wishes to reduce the total resistance. At a design operating speed of 23 kn, the only affine transformation in Figure 11 that showed promise was to increase the sectional beam-to-draft ratio, $B_1 / L$. This increase makes the sections closely semicircular, thereby reducing the frictional resistance.

However, a principal aim of this river vessel was to reduce the wave generation. In this regard, Figure 9a demonstrated that increasing the slenderness ratio by 50% resulted in a reduction of wavemaking at 23 kn of the order of 60%.

### 5.2. Future Studies

Further studies should include modifications of the demihulls in various ways. Suitable modifications include embracing the concept of pure semicircular sections. These will reduce the frictional resistance to the minimum. Secondly, it is possible that modifying the sectional-area curve (the longitudinal distribution of volume) may lead to a further reduction in wave generation.

Thirdly, if wave generation is a key design consideration, one should expand the current investigation into the rôle that the slenderness ratio plays. It is understood that the consquently very long hulls might lead to some practical operational difficulties. These difficulties include maneuverability in confined waterways and manning regulations, which might impose increased salaries for a larger crew.

**Funding:** This research received no external funding.

**Institutional Review Board Statement:** Not applicable.

**Informed Consent Statement:** Not applicable.

**Data Availability Statement:** Not applicable.

**Conflicts of Interest:** The authors declare no conflict of interest.

**List of Symbols**

| | | | |
|---|---|---|---|
| $A_T$ | Transom area | $T$ | Draft |
| $A_M$ | Midship-section area | $U$ | Ship velocity |
| $B$ | Overall beam | $W$ | Displacement weight |
| $B_1$ | Demihull beam | $b_1$ | Demihull local beam |
| $C_A$ | Correlation allowance | $d$ | Depth of water |
| $C_B$ | Block coefficient | $f_F$ | Frictional-resistance form factor |
| $C_P$ | Prismatic coefficient | $g$ | Acceleration due to gravity |
| $F$ | Froude number | $k$ | Circular wave number |
| $F_d$ | Depth Froude number | $k_x$ | Longitudinal wave number |
| $F_\nabla$ | Volumetric Froude number | $k_y$ | Transverse wave number |
| $L$ | Length | $s$ | Demihull centerplane separation |
| $L_P$ | Prototype nominal length | $w$ | Towing-tank or canal width |
| $L/\nabla^{1/3}$ | Slenderness ratio | $x$ | Longitudinal coordinate |
| $P$ | Power | $x_t$ | Longitudinal coordinate at transom |
| $R_A$ | Correlation resistance | $y$ | Transverse coordinate |
| $R_F$ | Frictional resistance | $z$ | Vertical coordinate |
| $R_H$ | Hydrostatic resistance | $\Delta$ | Model displacement mass |
| $R_T$ | Total resistance | $\Delta_P$ | Prototype displacement mass |
| $R_W$ | Wave resistance | $\zeta_t$ | Free-surface elevation on face of transom |
| $R_a$ | Aerodynamic resistance | $\eta$ | Propulsion efficiency |
| $R_N$ | Reynolds number | $\nu$ | Kinematic viscosity of water |
| $S$ | Overall wetted-surface area | $\rho$ | Density of water |
| $S_1$ | Demihull wetted-surface area | $\nabla$ | Displacement volume |

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
