# Peer review of "Reanalysis of the Sydney Harbor RiverCat Ferry"

_jmse, doi:10.3390/jmse9020215_

Round 1
Reviewer 1 Report
The paper is well written and is methodologically sound. The results are interesting and the figures support the argumentation. I would only expect some comments on the impact of shallow waters as well as the matching of the propellers with engine and the hull, incl. cavitation, noise, etc. issues.
Author Response
Reviewer 1:
(1) The paper is well written and is methodologically sound. The results are interesting and the figures support the argumentation.
Thank you.
(2) I would only expect some comments on the impact of shallow waters
I do not understand the comment of the reviewer requesting comments on “impact of shallow waters”, since indeed a major part of the paper is devoted this topic. For example, nearly all the equations include the effect of shallow water and the four parts of Figure 5 are devoted to this matter - particularly Figure 5(d).
(3) ... as well as the matching of the propellers with engine and the hull, incl. cavitation, noise, etc. issues.
This is a theoretical paper, devoted to wave resistance and total resistance. The paper is already 15 pages long and more practical matters are beyond the scope of the paper. However, I have added some suitable text at the end of Section 1.1. I have also added (new) Figure 3 and (new) Figure 4, which provide helpful details of the machinery.
Reviewer 2 Report
Interesting paper from the hydrodynamic and even multihull design point of view
- In figure 2, could you please add the propulsion? Place the propellers please
- It is the more slender hull I have never seen, this is the main reason why potential wave theory works so well. Have you compared your results with CFD results?
- Please provide a curve of the wave making resistance coefficient (Cw or Cr) for low Froude numbers, just to see the humps and valleys of this curve. Include Fn bellow 0.4 please.
- How did you arrive to the value of the form facto of line 157?
- In the graphics of page, there are different Rt/W curves, I guess the dots are experiments, but the other 2 curves?
- Could you please improve the conclusions sections? You have added some important comments in 4.3, 4.4 and 4.5
- How did you arrived to the propulsion efficiency of 0.621 of line 264?
Author Response
Reviewer 2:
(1) Interesting paper from the hydrodynamic and even multihull design point of view
Thank you.
(2) In figure 2, could you please add the propulsion? Place the propellers please
As noted for Referee 1, I have added some suitable text at the end of Section 1.1. I have also added (new) Figure 3 and (new) Figure 4, which provide helpful details of the machinery, including the propellers.
(3) It is the more slender hull I have never seen, this is the main reason why potential wave theory works so well. Have you compared your results with CFD results?
No, I have not compared the results with CFD. Doing so would greatly increase the scope and length of the paper.
(4) Please provide a curve of the wave making resistance coefficient (Cw or Cr) for low Froude numbers, just to see the humps and valleys of this curve. Include Fn bellow 0.4 please.
I am not sure about the advantage of providing such curves. There are many papers on this subject. I have added a couple of sentences referring the reader to such curves.
I should add here that the definition of the traditional wave-resistance coefficient and the traditional residuary-resistance coefficient, was based on the idea that typical ships of two centuries ago travelled at low speed so that most of the resistance was due to viscosity, and so the nondimensionalizing parameter \half\rho U^2 S used for the frictional or the viscous resistance was borrowed for the wave resistance and the residuary resistance. However, this is a poor choice for nondimensionalizing, because the wavemaking bears no physical relationship to \half\rho U^2 S.
Nevertheless, I have added some suitable text at the end of Section 4.1 and have referred the reader to a detailed discussion on this point.
(5) How did you arrive to the value of the form facto of line 157?
I have added a sentence explaining that a root-mean-square minimization scheme was used.
(6) In the graphics of page, there are different Rt/W curves, I guess the dots are experiments, but the other 2 curves?
It is standard engineering practice in presentation of results to use symbols (not dots) to indicate experimental data points and, on the other hand, curves to represent theoretical calculations. The description in the text also explains this. I do not understand why reviewer refers to "other 2 curves", when there are in fact 5 theoretical curves. The meaning of these curves is indicated clearly in the table inside the plot, in each case.
I have added a sentence clarifying the use of symbols to indicate the experimental data.
(7) Could you please improve the conclusions sections? You have added some important comments in 4.3, 4.4 and 4.5
If reviewer could indicate what he wants me to "improve", I will be happy to do so.
(8) How did you arrived to the propulsion efficiency of 0.621 of line 264?
I have clarified this in Section 4.6 after Equation 20.
Reviewer 3 Report
The paper is well prepared. And I have no comments.
Author Response
Reviewer 3:
(1) The paper is well prepared. And I have no comments.
Thank you.